# Investigating the relationship between teacher efficacy, job satisfaction, and digital resource utilization in assessment practices: Insights from PISA 2018 and 2022

Dirgha Raj Joshi[1], Jeevan Khanal[2*], Bishnu Maya Joshi[3]

**1** Mahendra Ratna Campus Tahachal, Tribhuvan University, Kathmandu, Nepal, **2** Nepal Open University, Lalitpur Metropolitan City, Nepal, **3** Department of Economics Education, Mahendra Ratna Campus Tahachal, Kathmandu, Nepal

\* jeevankhanal480@gmail.com

## Abstract

The COVID-19 pandemic disrupted global education systems, forcing rapid shifts in teaching practices, technology integration, and assessment methods. However, little is known about how teacher efficacy, job satisfaction, and digital adoption vary across economic contexts. Insufficient research examines how income levels influence these factors, hindering equitable support for educators in post-pandemic recovery. This study examines variations in teacher efficacy (TE), job satisfaction (JS), assessment practices (AP), and technology adoption (UIT/UDT) across low-, upper-middle-, and high-income countries (LMICs, UMICs, HICs) before and after the COVID-19 pandemic. Guided by Bandura's Social Cognitive Theory, the analysis explores how these factors interact and shift in response to pandemic-related disruptions in educational systems. The study utilizes PISA 2018 (pre-pandemic) and 2022 (post-pandemic) data from 128,866 teachers across 24 countries, employing structural equation modeling and machine learning as primary analytical techniques. Results indicate that job satisfaction significantly affects teacher efficacy but has minimal direct impact on the use of instructional technology tools. Teacher efficacy demonstrates a significant positive effect on both technology adoption (UIT) and assessment practices, while the use of digital learning and communication tools similarly influences assessment practices. These findings suggest that teacher efficacy and digital tool integration are key determinants of assessment practices. The study highlights how economic contexts shape teacher development, proposing targeted approaches for equitable post-pandemic education. HICs benefit from institutional support reinforcing the JS-TE relationship, while LMICs require solutions addressing resource gaps that impede consistent technology implementation. These evidence-based findings support context-specific policy interventions to enhance teacher support and digital integration globally.

**Data availability statement:** All relevant data are publicly available from the OECD Programme for International Student Assessment (PISA) 2018 and 2022 datasets (https://www.oecd.org/en/data/datasets/pisa-2022-database.html).".

**Funding:** The author(s) received no specific funding for this work.

**Competing interests:** The authors have declared that no competing interests exist.

## Introduction

The COVID-19 pandemic served as a global catalyst for digital education, forcing rapid adoption of instructional technologies (IT) and digital learning tools (DLC) across diverse economic contexts [1,2]. While these tools have become pedagogical necessities [3–5], their implementation reveals stark disparities rooted in teachers' self-efficacy (TE) and job satisfaction (JS) [6–8]. This study investigates how economic contexts moderate the relationships between TE, JS, and technology adoption (UIT/UDT) to shape assessment practices (AP), addressing critical gaps in understanding post-pandemic educational equity.

Pre-pandemic studies documented technology access gaps between HICs and LMICs [9] but neglected psychological adoption barriers. TE (teachers' belief in their instructional effectiveness) predicts pedagogical innovation [10], while JS mediates technology engagement through institutional support systems [11]. The pandemic exacerbated these divides: HICs maintained education continuity through robust infrastructure, whereas LMICs faced systemic collapse affecting 99% of students [12,13]. Despite global EdTech investments, the TE-JS-technology nexus remains underexplored in LMIC/UMIC contexts [14], particularly regarding assessment practices.

Bandura's (1986) Social Cognitive Theory (SCT) provides the foundational framework for this study, conceptualizing teacher efficacy (TE) as a personal factor, technology adoption (UIT/UDT) as behavioral dimensions, and school resources as environmental components within its triadic reciprocity model [15]. This framework explains the divergent outcomes observed across economic contexts: while teachers in high-income countries (HICs) leverage advanced tools like AI-driven assessments, their counterparts in low- and middle-income countries (LMICs) demonstrate remarkable agency by adapting low-resource solutions (e.g., SMS-based feedback) within constrained environments.

Three critical research gaps motivate this investigation. First, the empirical relationship between teacher efficacy (TE) and digital learning tool (DLC) adoption in assessment improvement remains unclear. Second, the moderating role of economic status on post-pandemic job satisfaction (JS)-technology engagement requires examination [16]. Third, the net impact of digital transitions on educational inequities – particularly concerning emerging AI risks (Capraro et al., 2024) – remains undetermined. While existing single-country studies (e.g., U.S. digital disparities [17] and Chinese contexts [18]) have explored these factors in isolation, they consistently overlook cross-contextual interactions between psychological factors (TE/JS), technological adoption (UIT/UDT), and economic conditions. This study addresses these gaps through a comparative analysis of 128,866 teachers across 24 countries using PISA 2018–2023 data. We operationalize Use of Instructional Technology (UIT) as the frequency of employing pedagogical tools (e.g., simulations, e-portfolios), while Use of Digital Learning Tools (UDT) measures the intensity of communication platform utilization (e.g., social media, spreadsheets). These precise definitions facilitate robust cross-country comparisons of classroom practices rather than technological innovation.

The investigation makes dual contributions: advancing theoretical understanding of SCT's boundary conditions in digital education contexts, while developing practical strategies for equitable technology integration aligned with Sustainable Development Goal 4 (SDG4) targets. By examining how macro-level economic contexts shape micro-level teacher behaviors during disruptions, the study offers novel insights for addressing persistent digital divides in global education systems. This study is guided by following research questions.

1. How do economic contexts (LMIC/UMIC/HIC) shape the interplay between TE, JS, and UTAUT/UDT, and to what extent did the COVID-19 pandemic amplify or mitigate these disparities?

2. How does TE influence AP through technology adoption (UTAUT/UDT), and what role do institutional resources play in mediating this relationship across economic groups?

## Literature review

### Overview of SCT and its relevance to the study

Albert Bandura's SCT (1986) offers a powerful framework for understanding the dynamic relationship between TE, JS, UIT/UDT, and AP across varying economic contexts [19]. At the core of SCT is the triadic reciprocity model, which integrates personal (cognitive beliefs), environmental (institutional and economic conditions), and behavioral (teaching practices) factors [20,21]. This model proves particularly valuable for analyzing teachers' pandemic responses, where resource and support system disparities produced markedly different outcomes between high-income countries (HICs) and low- to middle-income countries (LMICs/UMICs).

SCT identifies three primary sources of self-efficacy that shape teachers' ability to adapt: mastery experiences (e.g., successful technology use), vicarious learning (e.g., observing peers), and social support (e.g., institutional encouragement) [22]. The pandemic served as an environmental jolt that exposed deep educational inequities. While HIC teachers benefited from robust institutional support, including digital tool access, structured training, and collaborative networks that reinforced TE and sustained technology adoption [13], their LMIC/UMIC counterparts faced significant challenges. Infrastructure limitations (unreliable internet, scarce devices) and minimal professional development hindered efficacy-building, often forcing reversion to traditional assessment methods despite digital adoption pressures [23]. This divergence demonstrates how environmental constraints can override personal motivation, a dynamic SCT explains through its contextual moderators.

A critical application of SCT reveals literature gaps regarding economic context as a moderator of TE's impact. While TE strongly predicts technology use in well-resourced settings, its effects attenuate in LMICs due to systemic barriers [24]. For instance, high-efficacy teachers in these regions often cannot implement innovative practices because of infrastructural deficits. The pandemic's legacy further raises questions about sustainability: whereas HICs institutionalized remote-learning tools, LMICs achieved only temporary adaptations rather than systemic transformation [25]. Without targeted investments in training and infrastructure, these findings suggest post-pandemic gains may erode in resource-poor contexts.

SCT's focus on modifiable factors (e.g., peer collaboration, policy interventions) offers a strategic roadmap for addressing equity gaps. Research demonstrates that even in resource-constrained environments, teachers manifest agency through informal knowledge-sharing networks and incremental policy advocacy [26]. By employing SCT as our analytical framework, this study both explains global disparities in technology integration and identifies actionable interventions, including localized training programs and equitable resource allocation, to support teachers across economic contexts. SCT's unique emphasis on the person-environment-behavior interplay provides particularly valuable insights for navigating post-pandemic education systems' complexities.

### Teacher efficacy (TE) and technology integration

Teacher efficacy (TE), teachers' belief in their ability to effectively instruct and manage classrooms, has been widely studied as a critical factor influencing technology integration [27,28]. Research identifies several TE determinants: supportive

leadership, improved student outcomes, and peer collaboration enhance efficacy, while rigid teaching models and technological usability challenges diminish it [29]. Empirically, higher TE correlates with greater adoption of digital tools (UTAUT/UDT) and technology-enhanced assessment practices (AP) [30]. Bandura's SCT explains how mastery experiences and vicarious learning shape TE while revealing a paradox, teachers in under-resourced contexts often lack these efficacy-building opportunities, perpetuating cyclical disadvantage [31]. This necessitates policy interventions (e.g., simulated training, peer mentoring) in LMICs, where teachers consistently report lower TE than HIC counterparts due to resource and training disparities [32].

Contextual variations emerge: while leadership support and collaboration universally boost TE, their impact is mediated by environmental constraints. In LMICs, systemic barriers (e.g., scarce resources) often outweigh institutional support, demanding interventions that address both structural and individual factors [32]. The COVID-19 pandemic exposed these inequities, creating divergent trajectories: HICs teachers stabilized TE through institutional resources, whereas LMICs educators faced compounded stressors (device shortages, unreliable electricity), forcing survival-mode teaching over innovation [33]. These findings underscore the non-linear relationship between crisis response and sustained TE development.

## Job satisfaction (JS) and teacher performance

Job satisfaction (JS) has been extensively examined as a key predictor of teacher efficacy (TE) and technology adoption in education [34,35]. Research indicates that teachers with higher JS demonstrate greater instructional confidence and willingness to integrate digital tools, as professional fulfillment reinforces motivation [36]. Key questions in this domain examine JS variations across contexts and its direct influence on TE. Institutional support and workload emerge as critical JS determinants, where inadequate resources or excessive administrative demands reduce satisfaction [37].

The JS-TE relationship shows contextual dependencies: in HICs, institutional support buffers job demands, allowing JS to directly enhance TE, while in LMICs, systemic constraints often decouple satisfaction from efficacy without targeted resource allocation [38]. JS mediates between self-efficacy, life satisfaction, and workload, highlighting its role in teacher well-being [39]. This mediation is economically contingent—LMIC teachers require both policy changes and infrastructural investments to convert reforms into efficacy gains, a nuance absent from universal models. Collectively, these findings emphasize fostering JS through supportive environments, equitable resources, and needs-based policies to enhance both teaching performance and technology integration.

## Digital tools adoption (UIT/UDT) in education

The adoption of digital tools in education (UIT/UDT) is strongly tied to teachers' self-efficacy beliefs, a central component of Bandura's SCT [10]. While prior research confirms that efficacy drives technology use [22], a deeper synthesis reveals contextual disparities: Al-Adwan et al. [40] demonstrate that Technological, Pedagogical, and Content Knowledge (TPACK) enhances self-efficacy and adoption only when institutional support mitigates technostress and resistance to change, a nuance frequently absent from HIC studies. SCT's framework clarifies this relationship by identifying mastery experiences and vicarious learning as efficacy sources [41], yet post-pandemic findings [42,43] show economic contexts fundamentally moderate these mechanisms. LMIC teachers faced compounded barriers (e.g., resource gaps) that eroded efficacy despite high motivation, while HIC educators leveraged institutional support to sustain blended learning transitions [44].

The literature critically lacks comparative analysis of SCT's agency-context interplay across economic settings. While studies [45,46] present training as a universal solution, research [47] reveals UIT/UDT's impact on assessment practices diverges sharply in LMICs due to infrastructural constraints. This gap demands context-specific interventions: where HICs benefit from systemic efficacy reinforcement (e.g., peer modeling), LMICs require structural solutions (e.g., equitable access) to bridge SCT's theoretical assumptions with on-the-ground realities.

## Assessment practices (AP) in the digital era

The digital transformation of assessment practices reveals a dynamic SCT-framed interplay between teacher efficacy (TE), technology adoption (UIT/UDT), and pedagogical outcomes [48]. SCT's triadic reciprocity model elucidates how teachers' behavioral factors interact with environmental conditions to shape assessment quality, particularly in digital tool integration [25]. Research documents significant methodological variations, showing how technology enables innovative assessment approaches that address diverse learning needs and improve outcomes [49].

TE directly influences assessment practices, with higher-efficacy teachers demonstrating more innovative digital tool use [50]. Social contagion processes further strengthen this relationship, as colleagues' practices measurably influence teachers [51,52]. Emerging technologies like AI-driven reflective assessment tools create new TE-enhancement opportunities through data-driven insights [53], with effective tool use mediating between TE and assessment innovation [54].

Yet these advancements remain unevenly distributed. Persistent equity gaps emerge in resource-constrained settings where limited technology access or training creates implementation barriers [55,56]. SCT's dual focus on contextual barriers and teacher agency provides a critical framework for these disparities, showing how environmental factors constrain even highly efficacious teachers. The literature confirms that while technology transforms assessment paradigms, equitable implementation requires both TE development and systemic support addressing access inequities.

## Economic context as a moderator

Bandura's SCT [15] offers a critical framework for analyzing how economic contexts moderate educational technology integration by shaping teacher agency, self-efficacy (TE), and environmental constraints [57]. At the macro level, national income disparities (LMICs/HICs) create differential environmental conditions within SCT's triadic reciprocity model: resource scarcity in LMICs restricts digital tool access, limiting mastery experiences crucial for developing technology-related TE [58]. Cross-country adoption patterns reflect this dynamic, HIC teachers benefit from abundant vicarious learning opportunities (observing peer implementations) and institutional support that reinforce efficacy beliefs, including formal professional development and mentorship programs [59]. These environmental advantages foster stronger TE by providing positive experiences and support systems [60].

Conversely, LMIC economic barriers create disempowering conditions that undermine TE despite policy mandates promoting technology use. The pandemic exemplified this SCT-aligned pattern: well-resourced systems enabled teachers to convert crisis-induced changes into mastery opportunities, while impoverished systems' environmental constraints eroded efficacy [61]. SCT thus reveals economic context as an active moderator, not mere backdrop—of teachers' behavioral capacity for technology adoption, determining whether environments provide efficacy-developing or efficacy-inhibiting conditions [45]. This theoretical lens confirms that equitable technology integration requires interventions addressing both individual TE and environmental affordances.

## Pandemic as a disruptive environmental factor

The COVID-19 pandemic functioned as a profound environmental jolt (SCT) that disrupted global education systems [62], fundamentally altering TE and technology adoption patterns. Bandura's triadic reciprocity model elucidates the crisis-time interactions between environmental (lockdowns), personal (TE), and behavioral (technology use) factors [63]. Although emergency remote teaching initially undermined TE through inadequate mastery experiences [64,65], some educators utilized institutional support to build resilience via trial-and-error experimentation and peer learning, ultimately accelerating technology adoption [66]. Stark economic disparities emerged, with HICs institutionalizing long-term changes through sustained resource access, while LMICs grappled with persistent challenges like device shortages [13]. Teacher communities' adaptive strategies showcased collective efficacy in overcoming environmental barriers [67,68], validating SCT's perspective on crises as transformation catalysts.

Critical literature gaps persist. First, insufficient longitudinal, cross-country comparisons of pandemic impacts limit our understanding of recovery trajectories. Second, the mechanisms differentiating technology adoption (UIT/UDT) between LMICs and HICs remain unclear. Third, contextual variations in TE's effect on assessment practices (AP) and digital tool use (UDT) require further exploration. Fourth, economic moderators of the job satisfaction (JS)-TE relationship are under-studied, particularly in resource-constrained settings. Fifth, technology thresholds for effective TE-AP mediation in LMICs remain undefined. Addressing these gaps demands longitudinal mixed-methods designs and policy-intervention studies to develop context-specific implementation strategies that align with SCT's person-environment-behavior framework.

### Present study

This study employs Bandura's [15] SCT as its conceptual foundation to examine how TE, JS, AP, and technology adoption (UTAUT/UDT) interact across different economic contexts (LMICs, UMICs, and HICs) before and after the COVID-19 pandemic. Grounded in SCT's principle of triadic reciprocity, we hypothesize that economic contexts will significantly moderate the relationships between TE, JS, and technology adoption, with pandemic disruptions amplifying existing disparities due to varying institutional resources. The framework further proposes that TE will positively influence AP through technology adoption, as efficacy beliefs drive behavioral outcomes [69], while JS will reinforce TE's impact on technology use, particularly in well-resourced environments [70].

By analyzing pre- and post-pandemic data, the study specifically investigates: (1) how economic disparities shape the TE-JS-UTAUT/UDT interplay; (2) the mediating role of technology adoption in the TE-AP relationship; and (3) the boundary conditions created by institutional resources. This approach not only tests SCT's universal applicability but also reveals context-specific patterns where environmental constraints may override personal factors in LMICs [55].

The conceptual framework (Fig 1) thus provides a comprehensive lens to understand how teachers' capabilities, motivations, and technology behaviors interact within environmental constraints to shape post-pandemic educational outcomes, while offering actionable insights for equitable policy interventions across economic contexts.

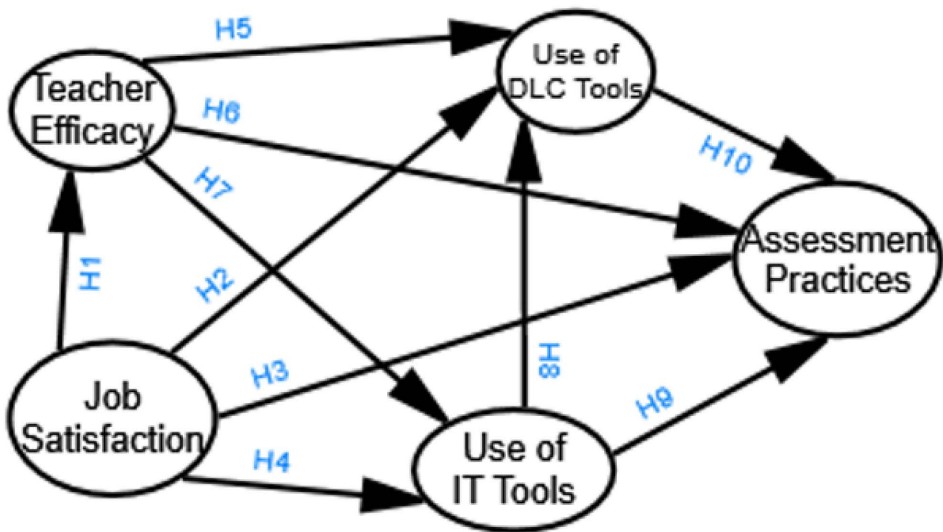

**Fig 1. Conceptual Framework.**

## Methodology

This study is based on data from the Programme for International Student Assessment (PISA) for 2018 (pre-pandemic) and 2022 (post-pandemic). A total of 128,866 teachers participated, with 48.6% from the pre-pandemic period and 51.4% from the post-pandemic period, representing 24 countries worldwide. Countries were categorized by income level according to the World Bank's 2022–2023 classifications (https://tinyurl.com/4phrxwbv) into three groups: lower-middle-income (LMIC; $1,136–$4,465), upper-middle-income (UMIC; $4,466–$13,845), and high-income (HIC; ≥ $13,846). Low-income countries were excluded due to data unavailability. The sample included Morocco (LMIC); Albania, Azerbaijan (Baku), Brazil, Colombia, Costa Rica, the Dominican Republic, Georgia, Kosovo, Malaysia, and Peru (UMIC); and Australia, Chile, Chinese Taipei, Germany, Hong Kong, Korea, Macao, Panama, Portugal, Spain, the UAE, the UK, and the USA (HIC) [13].

This study analyzes publicly available secondary data from the OECD's PISA 2018 and 2022 teacher questionnaires (https://tinyurl.com/y6am76n5), accessible via the OECD website (https://www.oecd.org/pisa/). These standardized surveys, administered by the OECD and national coordinators, collected responses on teacher efficacy (TE), job satisfaction (JS), assessment practices (AP), and technology adoption (UIT/UDT). The structured questionnaire design (e.g., items TC018Q01N–TC018Q11N) captures subject-specific training and teaching practices with operational clarity (e.g., help-button definitions for "Modern foreign languages"). The OECD ensures cross-national comparability through rigorous protocols, including pilot-tested translations, standardized scoring, and data-cleaning mechanisms (e.g., consistency checks for unmarked responses). Prior OECD technical reports validate the questionnaires' reliability for measuring constructs like JS and AP. While this study uses secondary data, the original surveys serve as primary sources for teacher respondents, ensuring data authenticity while enabling robust secondary analysis.

### Research instrument

From the PISA database, thirty-one items across five constructs were included in the study, with construct validity confirmed through confirmatory factor analysis (CFA). The five constructs were: teacher efficacy (TE), job satisfaction (JS), assessment practices (AP), use of instructional technology tools (UIT), and use of digital learning and communication tools (UDT). The number of items and measurement scales for each construct are detailed in Table 1.

**Teacher efficacy (TE).** Teachers' efficacy represents the ability of teachers to get students to believe they can do well in school work (TE01), help students value learning (TE02), craft good questions for the students (TE03), motivate students who show low interest in school work (TE04), and help students think critically (TE05) in their teaching.

**Job satisfaction (JS).** Job satisfaction represents the level of agree of teachers on the advantages of being a teacher clearly outweigh the disadvantages (JS01), choose to work as a teacher (JS02), regret that to decide to become a teacher (JS03), enjoy working at this school (JS04), wonder whether it would have been better to choose another profession (JS05), would recommend my school as a good place to work (JS06), satisfied with my performance in this school (JS07), and all in all, satisfied with my job (JS08).

**Assessment practices (AP).** Assessment practices represent the frequency of teachers for assessing student learning in having individual students answer questions in front of the class (AP01), provide written feedback on student work in addition to a mark (AP02), let students judge their own progress (AP03), observe students when working and provide immediate feedback (AP04), and collect data from classroom assignments or home work (AP05).

**Use of instructional technology tools (UIT).** Use of instructional technology tools represents the frequency of using data logging and monitoring tools (UIT1), simulations and modelling software (UIT2), e-portfolios (UIT3), multimedia production tools (UIT4), concept mapping software (e.g., Inspiration, Webspiration) (UIT5), and graphing or drawing software (UIT6) in their instructional practices.

**Table 1. Details of number of items and measurement scale under all constructs.**

| Constructs | No. of items | Measurement scale | Item codes in the *Teacher questionnaire* of PISA |
|---|---|---|---|
| Teacher efficacy (TE) | 5 | Not at all, to some extent, quite a bit, and a lot | TC199Q01HA-TC199Q07HA, excluding TC199Q04HA, and TC199Q06HA |
| Job satisfaction (JS) | 8 | Strongly disagree, disagree, agree, and strongly agree | TC198Q01HA-TC198Q10HA excluding TC198Q03HA (not in the item list and database) and TC198Q08HA |
| Assessment practices (AP) | 5 | Never or almost never, some lessons, most lessons, and every lesson or almost every lesson | TC054Q03NA- TC054Q07NA |
| Use of instructional technology tools (UIT) | 6 | Never, in some lessons, in most lessons, and in every or almost every lesson | TC169Q05HA- TC169Q05HA and TC169Q13HA- TC169Q14HA |
| Use of digital learning and communication tools (UDT) | 7 | | TC169Q01HA - TC169Q04HA and TC169Q09HA- TC169Q12HA |

*Source*: https://www.oecd.org/en/data/datasets/pisa-2022-database.html & https://tinyurl.com/yc62pz9c.

**Use of digital learning & communication tools (UDT).** Use of digital learning and communication tools represents the frequency of using word-processors or presentation software (UDT1), social media (e.g., Facebook, Twitter) (UDT2), communication software (e.g., email, blogs) (UDT3), computer-based information resources (UDT4), tutorial software or practice programmes (UDT5), digital learning games (UDT6), and spreadsheets (e.g., Microsoft Excel) (UDT7)

**Reliability and validity.** Table 2 assess the reliability and validity of the instrument used in the study. Composite reliability (CR) values for all constructs range from 0.75 to 0.86, indicating acceptable internal consistency [71,72]. The average variance extracted (AVE) values are found to be 0.37 to 0.56 suggesting that the constructs are acceptable [73]. The heterotrait-monotrait (HTMT) analysis shows correlations between constructs and all values are less than the threshold value of 0.90 [73] suggesting that the discriminant validity existed.

## Data analysis

Two teacher's databases of PISA (File named "CY07_MSU_TCH_QQQ" and "CY08MSP_TCH_QQQ") were used in the research. Both databases were merged with reference to the common variables in both database which were used in this research. Data was cleaned before analysis. The research utilized several major statistical techniques, including mean, standard deviation (SD), and structural equation modeling (SEM). The mean was used to assess the status of items before and after the COVID-19 pandemic, as well as across different economic levels of countries, with the mean values presented in a bar chart. Similarly, Flexplot were used to show the distribution of data as well as relationship between the variables teacher efficacy (TE), job satisfaction (JS), assessment practices (AP), use of instructional technology (IT) tools (UIT), and use of digital learning and communication (DLC) tools (UDT) based on BACovid and CLE. Furthermore, SEM was used to examine the results of hypothesized model as impact of teacher efficacy, job satisfaction, use of instructional

**Table 2. Reliability and validity of the instrument.**

| Constructs | CR | AVE | HTMT Analysis | | | | |
|---|---|---|---|---|---|---|---|
| | | | TE | JS | AP | UIT | UDT |
| Teacher efficacy (TE) | 0.86 | 0.56 | | | | | |
| Job satisfaction (JS) | 0.85 | 0.41 | 0.40 | | | | |
| Assessment practices (AP) | 0.75 | 0.38 | 0.51 | 0.28 | | | |
| Use of instructional technology tools (UIT) | 0.84 | 0.47 | 0.32 | 0.14 | 0.44 | | |
| Use of digital learning & communication tools (UDT) | 0.81 | 0.37 | 0.30 | 0.15 | 0.47 | 0.89 | |

technology tools, and use of digital learning and communication tools on assessment practices. Moreover, regularized linear regression under machine learning was applied to find the item wise effect of the variables with reference to hypothesized model.

## Results

Fig 2 presents the distribution of mean scores, while Fig 4 displays the standard deviations for key constructs, teacher efficacy (TE), job satisfaction (JS), assessment practices (AP), use of instructional technology (UIT), and use of digital learning and communication tools (UDT), across economic groups and time periods (pre- and post-COVID-19 pandemic). The results reveal significant variations in these measures among lower-middle-income (LMC), upper-middle-income (UMC), and high-income (HIC) countries, as well as between the pre-pandemic (BCP-2019) and post-pandemic (ACP-2023) phases.

Fig 3 shows that mean scores tend to be higher in HICs compared to LMCs and UMCs, indicating that high-income countries generally exhibit greater teacher efficacy, job satisfaction, technology use, and assessment practices. The ACP (after COVID-19 pandemic) category also shows higher scores than BCP (before COVID-19 pandemic), suggesting that the pandemic may have accelerated the adoption of instructional technology (UIT) and digital learning tools (UDT) across all income levels. However, the variations among country groups indicate that lower-income countries may still face challenges in integrating these tools effectively. Fig 4 displays the distribution of standard deviations, which reflect the variability of responses across different groups. Higher standard deviations in LMCs and UMCs suggest greater disparities in teacher efficacy, job satisfaction, and technology adoption within these countries, whereas HICs show more consistency in their responses. The standard deviations for UIT and UDT are relatively higher post-pandemic, implying that while technology adoption increased, its impact varied significantly among individuals and institutions.

Fig 4a–4f illustrates the distribution of teacher efficacy (TE), job satisfaction (JS), use of digital tools (UDT, UIT), and assessment practices (AP) across different economic categories (LMIC, UMIC, HIC) before and after the COVID-19 pandemic (BCP, ACP). Fig 4a shows that higher TE and JS are positively associated with AP, with

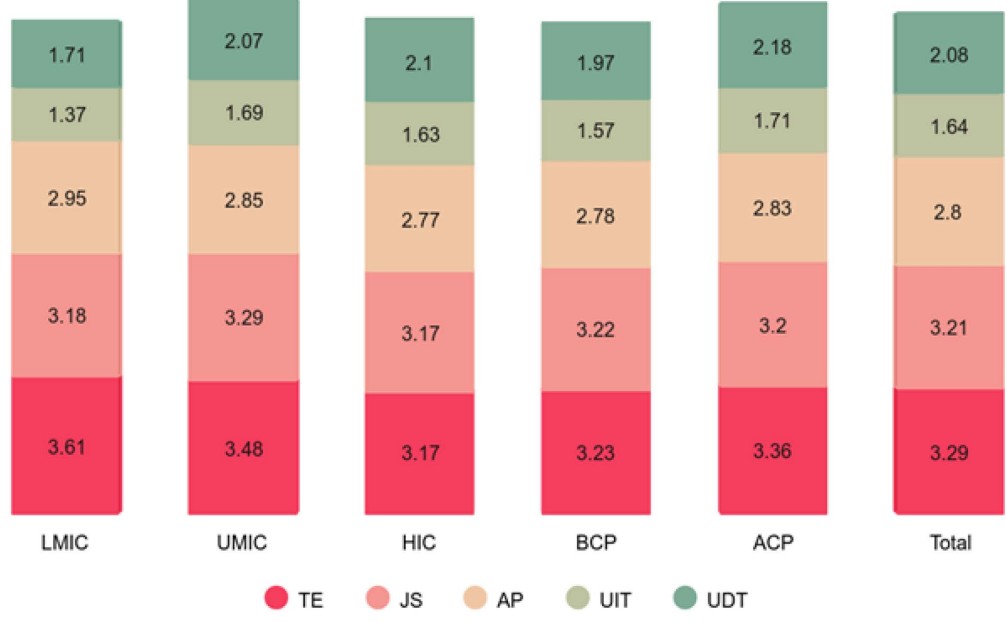

**Fig 2. Distribution of mean score.**

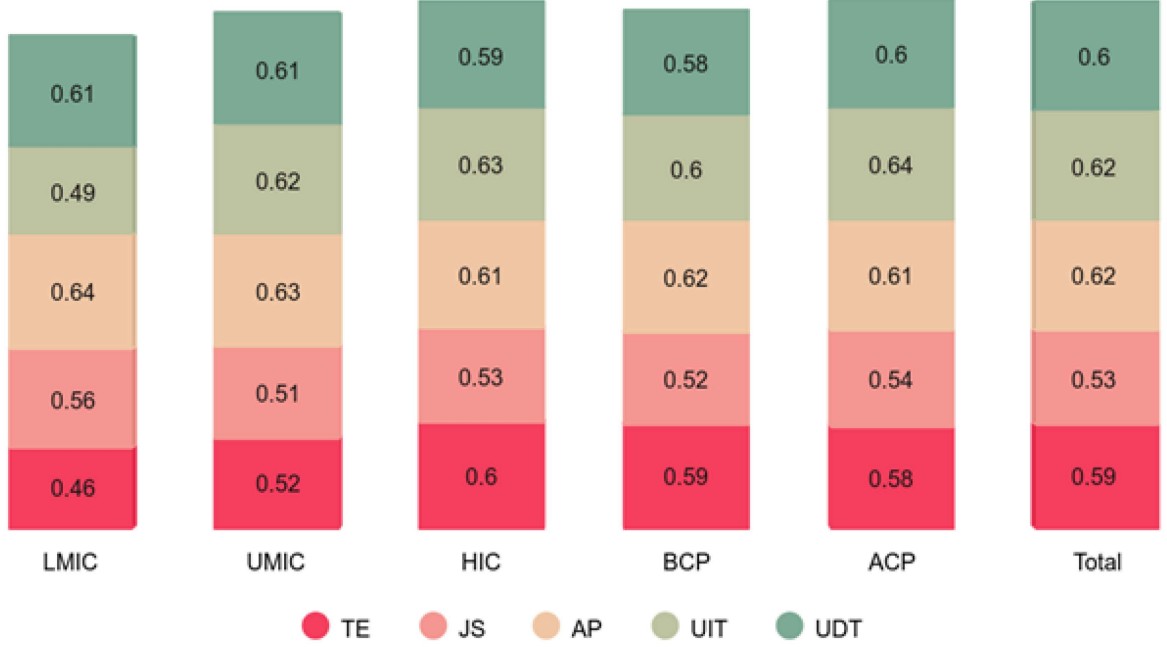

**Fig 3. Distribution of standard deviation.** LMIC-Lower middle-income countries; UMIC-Upper middle-income countries; HIC-High-income countries; BCP-Before COVID pandemic (2018); ACP-After COVID pandemic (2022); TE-Teacher efficacy; JS-Job satisfaction; AP-Assessment practices; UIT- Use of instructional technology (IT) tools; UDT- Use of digital learning & communication (DLC) tools.

variations across economic groups. Fig 4b suggests that UDT and UIT influence AP differently, with a stronger impact observed in HICs. Fig 4c and 4d indicate that TE and JS play a role in UDT and UIT adoption, with noticeable shifts ACP. The findings highlight that economic status and pandemic disruptions influenced teachers' instructional and assessment practices, with digital tools being more prevalent in wealthier nations ACP. Fig 4e shows a positive relationship between job satisfaction (JS) and teacher efficacy (TE) across economic categories, with a stronger correlation in high-income countries (HIC). Fig 4f illustrates the distribution of JS before and after COVID-19, revealing slight variations across economic groups. Overall, JS appears more stable in wealthier nations, with greater variability in lower-income countries.

## Results of structural equation modeling

The model of Fig 5 shows that the value of different model fit indices as Normed Fit Index (NFI), Relative Fit Index (RFI), Incremental Fit Index (IFI), Tucker-Lewis Index (TLI), and Comparative Fit Index (CFI) all range between 0.86 and 0.88, suggesting an acceptable but not excellent fit (values closer to 0.90 or above typically indicate a better fit) (Dion, 2008; Lam, 2012; [9,73,74]& Malekmohammadi, 2013). The Root Mean Square Error of Approximation (RMSEA) is 0.06, which indicates a reasonable fit (values below 0.08 suggest a good fit), meaning the model is likely an adequate representation of the data but could be improved [75,76].

The structural equation model (SEM) in the figure illustrates the relationships between teacher efficacy, job satisfaction, the use of digital learning and communication (DLC) tools, IT tools, and assessment practices. The results indicate that teacher efficacy positively influences job satisfaction (0.40), suggesting that teachers who feel more competent tend to be more satisfied with their jobs. Job satisfaction, in turn, has a strong impact on the use of both IT tools (0.50) and DLC tools (0.47), highlighting that satisfied teacher are more likely to integrate technology into their teaching practices.

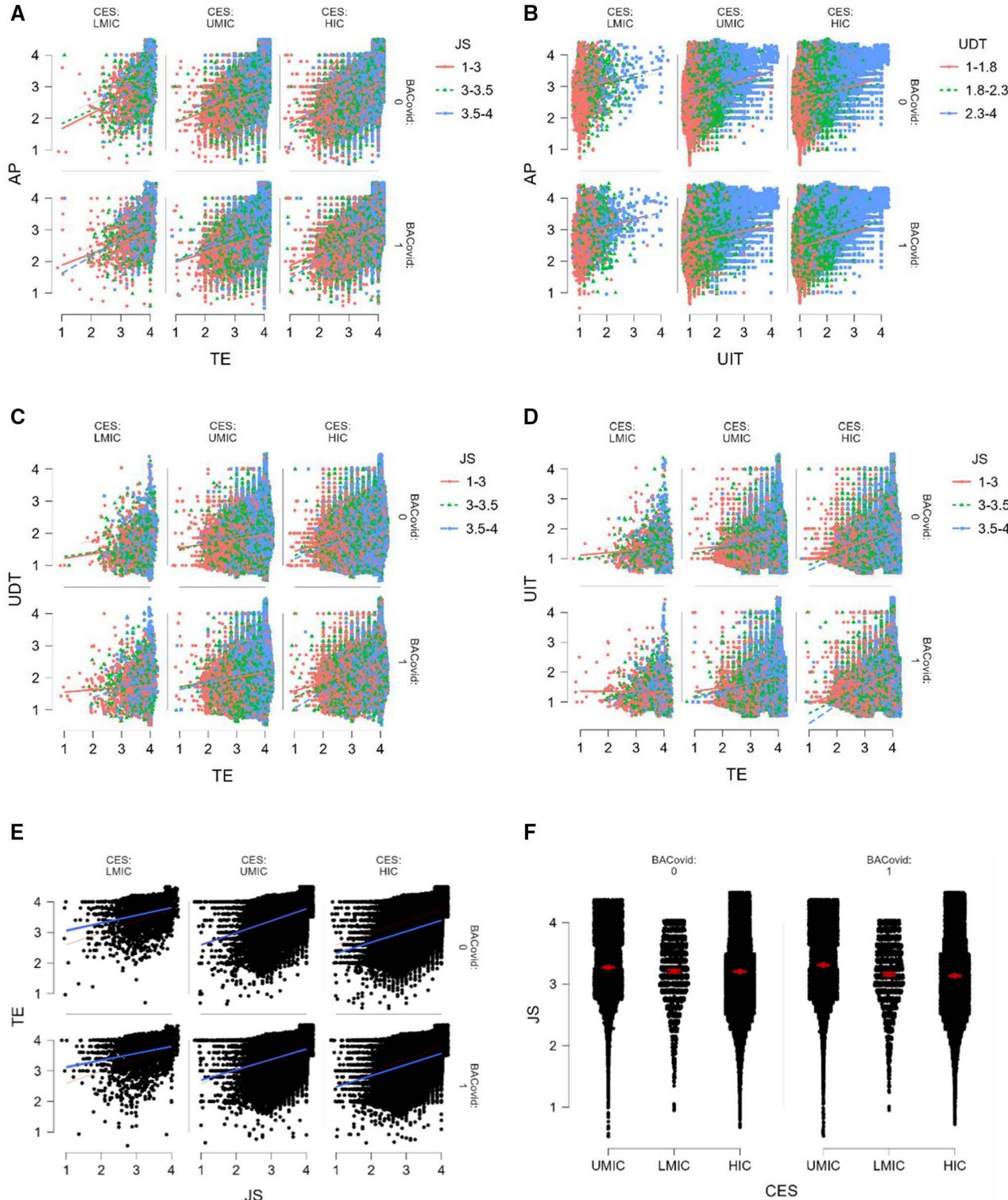

**Fig 4. a. Distribution of the effect of job satisfaction and teacher efficacy on assessment practices among different categories of country economy and BAP. b.** Distribution of the effect of UDT and UIT on assessment practices among different categories of country economy and BAP. **c.** Distribution of the effect of TE and JS on UDT among different categories of country economy and BAP. **d.** Distribution of the effect of TE and JS on UIT

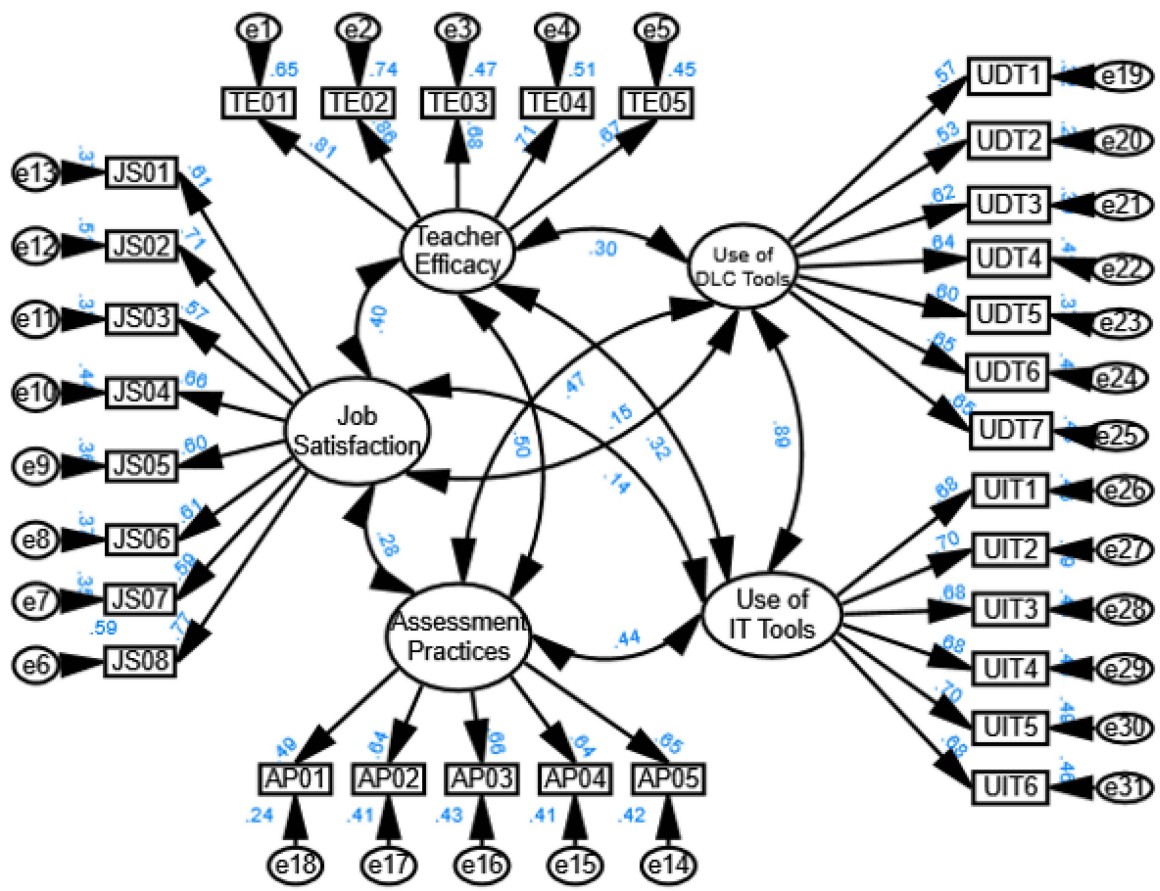

**Fig 5. Details of factor loadings.**

Furthermore, the use of IT tools significantly predicts assessment practices (0.44), showing that technology adoption enhances assessment methods. The use of DLC tools also positively impacts assessment practices (0.32) and IT tool usage (0.88), reinforcing the role of digital tools in modern education. Overall, the findings suggest that enhancing teacher efficacy and job satisfaction can lead to greater technology adoption, which ultimately improves assessment practices. This highlights the importance of professional development and workplace support in promoting effective digital learning strategies.

The model of Fig 6 shows that the value of different model fit indices as Normed Fit Index (NFI) is 0.88, Relative Fit Index (RFI) is o.86, Incremental Fit Index (IFI) is 0.88, Tucker-Lewis Index (TLI) is 0.86, and Comparative Fit Index (CFI) is 0.88, suggesting an acceptable because the values closer to 0.90 or above typically indicate a better fit [9,73,74]. The Root Mean Square Error of Approximation (RMSEA) is 0.06, which indicates a reasonable fit [9,75,76]..

Fig 6 presents a structural equation model (SEM) illustrating the relationships among job satisfaction (JS), teacher efficacy (TE), use of instructional technology (UIT), use of digital learning and communication (UDT), and assessment

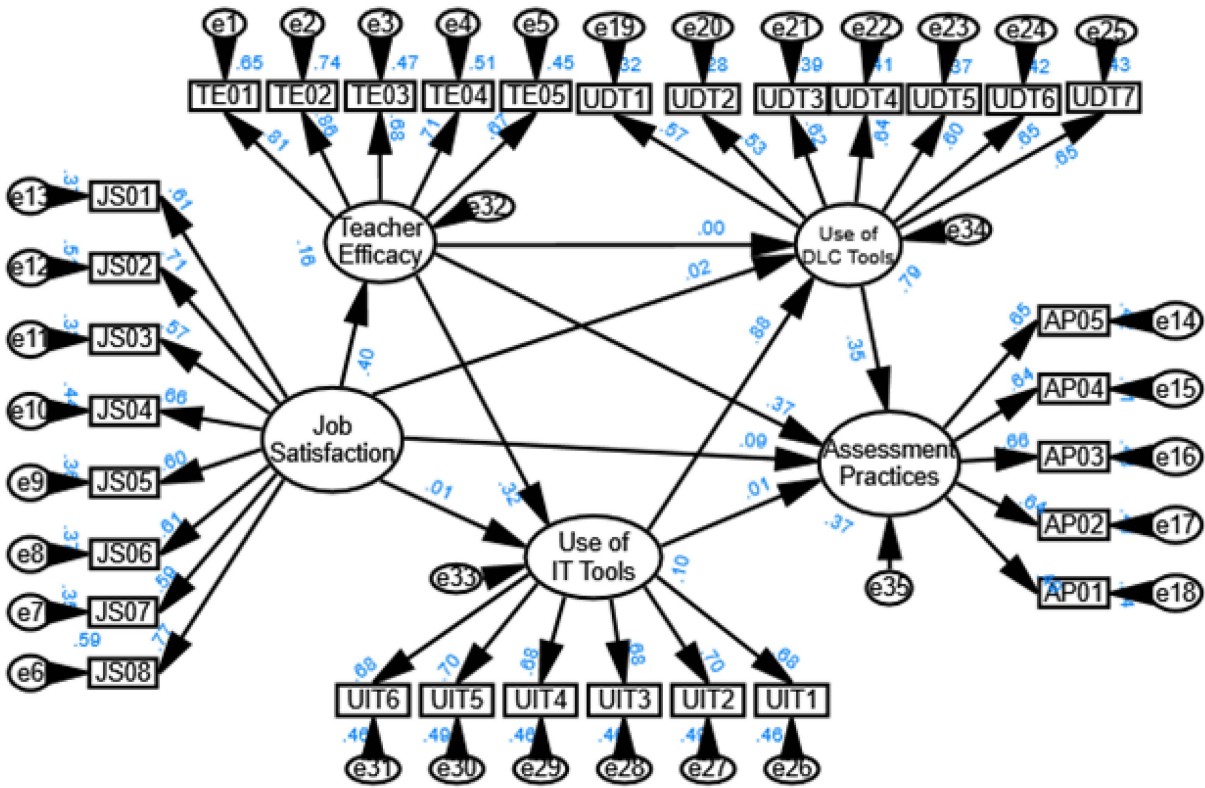

**Fig 6. Results of hypothesis of hypothesized model.**

practices (AP). The model explaining 16% variance in teacher efficacy, 10% in use of IT tools, 79% in the use of DLC tools, and 39% in assessment practice. The standardized path coefficients indicate the strength and direction of these relationships. JS positively influences TE (Beta = 0.40), suggesting that more satisfied teachers tend to exhibit higher efficacy. However, JS has a weaker effect on UIT (Beta = −0.32) and UDT (Beta = 0.01), indicating that job satisfaction does not strongly drive technology use. TE significantly predicts AP (Beta = 0.37), meaning that teachers with higher efficacy are more likely to adopt diverse assessment strategies. Additionally, TE has a strong effect on UDT (Beta = 0.88), suggesting that self-efficacious teachers are more inclined to use digital communication tools. UIT and UDT show moderate positive effects on AP (Beta = 0.37 and Beta = 0.35, respectively), highlighting the role of technology integration in assessment strategies. Interestingly, the relationship between JS and AP is weak (Beta = 0.01), implying that job satisfaction does not directly influence assessment practices. These results suggest that while teacher efficacy is a key driver of technology use and assessment methods, job satisfaction primarily influences efficacy rather than instructional practices.

## Results of machine learning

The model performance metrics in Table 3 provide insights into how well the hypothesized model predicts various factors influencing educational technology and teacher efficacy. The mean squared error (MSE) values indicate that the model achieves the lowest error in predicting the effect on the use of DLC tools (MSE = 0.16) and the highest error in predicting the effect on the use of IT tools (MSE = 0.35). The scaled MSE follows a similar pattern, with the highest value (1.37) for IT tools and the lowest (0.52) for DLC tools. The root mean squared error (RMSE) and mean absolute error (MAE) show that prediction accuracy is slightly better for DLC tools (RMSE = 0.40, MAE = 0.31) compared to other categories. The

**Table 3. Model Performance Metrics based on item wise effect in hypothesized model.**

| Indicators | Effect on assessment practice | Effect on the use of DLC tools | Effect on the use of IT tools | Effect on teacher efficacy |
|---|---|---|---|---|
| MSE | 0.28 | 0.16 | 0.35 | 0.30 |
| MSE (scaled) | 0.95 | 0.52 | 1.37 | 1.23 |
| RMSE | 0.52 | 0.40 | 0.59 | 0.54 |
| MAE/ MAD | 0.42 | 0.31 | 0.46 | 0.45 |
| MAPE | 16.50% | 16.74% | 29.26% | 15.04% |
| R² | 0.27 | 0.55 | 0.10 | 0.15 |
| λ | $4.212 \times 10^{-4}$ | $5.305 \times 10^{-4}$ | $3.157 \times 10^{-4}$ | $5.704 \times 10^{-4}$ |
| n(Train) | 65188 | 67340 | 69600 | 70771 |
| n(Validation) | 16297 | 16836 | 17400 | 17693 |
| n(Test) | 20371 | 21043 | 21750 | 22115 |
| Validation MSE | 0.28 | 0.16 | 0.35 | 0.30 |
| Test MSE | 0.28 | 0.16 | 0.35 | 0.30 |

mean absolute percentage error (MAPE) is highest for IT tools (29.26%), indicating greater variability in predictions, while it is lowest for teacher efficacy (15.04%). The R² values suggest that the model explains 55% of the variance in DLC tool usage but only 10% in IT tool usage. The lambda (λ) regularization values (Fig 7e–7h) are relatively similar across models. Training, validation, and test dataset sizes are consistent, ensuring reliability.

The regression models depict how various predictors contribute to different educational outcomes. In the first equation (Fig 7a), AP is modeled as a function of teacher-related indicators (TE01–TE05), job satisfaction variables (JS01–JS08), and measures of technology use (UIT and UDT). Each coefficient indicates the expected change in AP for a one-unit increase in the corresponding predictor, with TE03 (0.063) and TE04 (0.059) showing relatively stronger positive associations, suggesting that specific teacher attributes significantly boost AP. The second equation (Fig 7b) predicts UDT, where the intercept is 2.064 and predictors include teacher and job satisfaction variables, alongside IT usage measures. Notably, UIT4 has a high coefficient (0.18), indicating a robust positive influence on UDT. The third model (Fig 7c) relates UIT to teacher and job satisfaction indicators; while most coefficients are positive, the negative coefficient for JS03 (–0.062) reveals that some job satisfaction aspects might reduce UIT. Lastly, the fourth equation (Fig 7d) shows TE being influenced solely by job satisfaction variables, with JS07 (0.124) and JS08 (0.065) having the most pronounced positive impacts on teacher efficacy. Overall, these equations illuminate the intertwined effects of teacher characteristics, job satisfaction, and technology use on various educational performance metrics.

## Discussion

Our findings advance theoretical understanding of teacher behaviors by both supporting and challenging SCT [15], while revealing key connections to other frameworks. While SCT explains the triadic reciprocity between personal factors (TE/JS), behaviors (UIT/UDT), and environmental contexts (economic conditions), our results show important limits to its universal use. The higher scores in HICs (TE, JS, UIT, UDT, AP) support SCT's focus on environmental effects [22], but also show where the theory needs adjustment – especially for LMIC cases where strong TE didn't lead to technology adoption, a pattern better explained by technology acceptance models.

The pandemic's uneven impact across economic contexts yields key theoretical insights. While global digital adoption trends align with SCT's environmental disruption concept [77], the greater variability in LMICs/UMICs suggests Self-Determination Theory [78] may offer complementary explanation, where basic resource access precedes teachers' ability to actualize efficacy beliefs. This challenges SCT's universalist assumptions [79] while supporting Guillaume et al.'s [32] claim that economic context fundamentally reshapes motivational frameworks. Notably, the study doesn't argue the

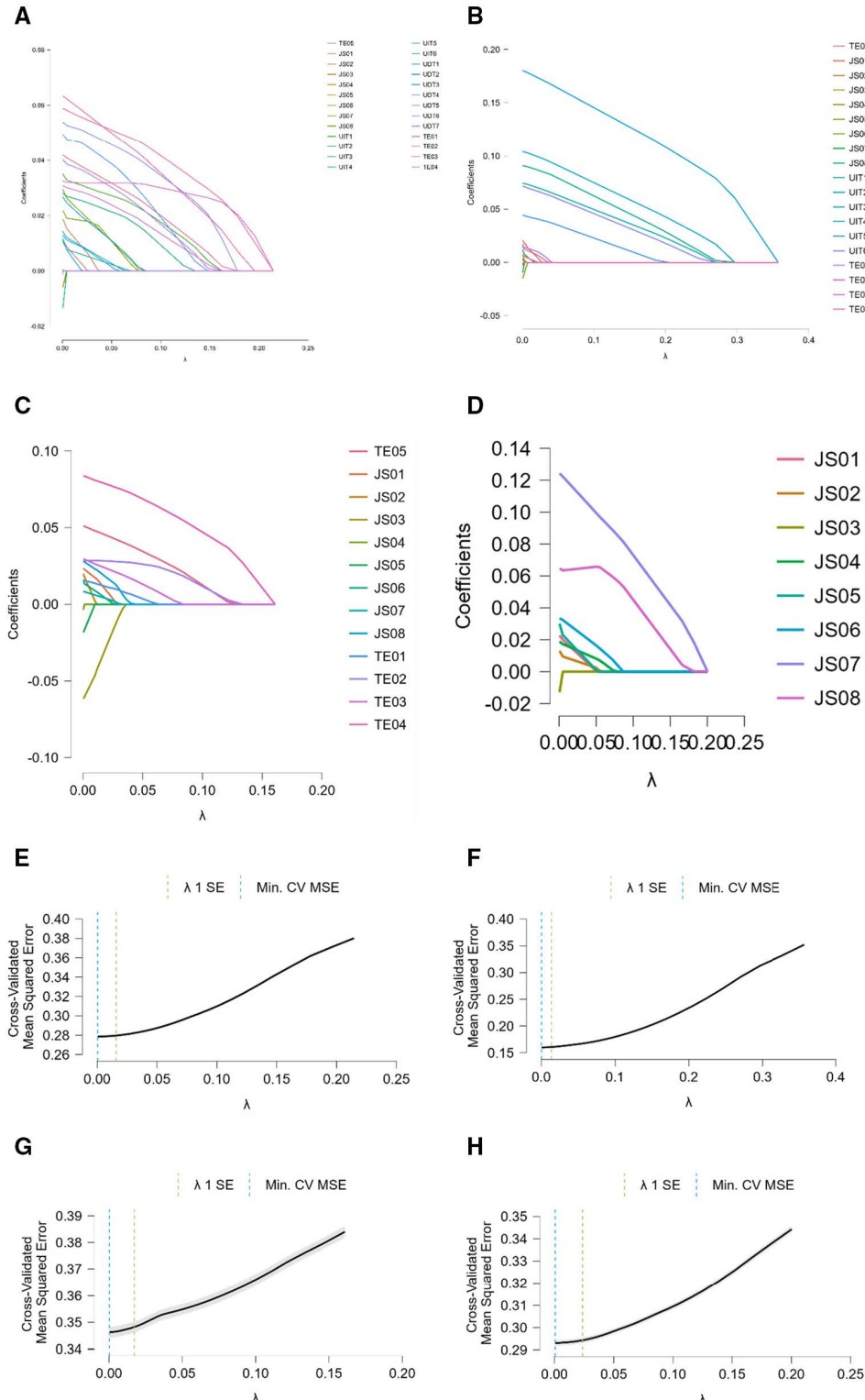

**Fig 7. a. Validation Trace Plot of assessment practice.** AP = 2.799 + 0.031 x TE01 + 0.033 x TE02 + 0.063 x TE03 + 0.059 x TE04 + 0.042 x TE05 + 0.011 x JS01 + 0.019 x JS02 + 0.001 x JS03 - 0.006 x JS04 + 0 x JS05 - 0.006 x JS06 + 0.03 x JS07 + 0.022 x JS08 + 0.035 x UIT1 - 0.001 x UIT2 + 0.028 x UIT3 - 0.014 x UIT4 + 0.011 x UIT5 + 0.015 x UIT6 + 0.012 x UDT1 + 0.027 x UDT2 + 0.013 x UDT3 + 0.049 x UDT4 + 0.054 x UDT5 + 0.04 x

UDT6 - 0.001 x UDT7. **b.** Validation Trace Plot of UDT. UDT = 2.064 + 0.017 x TE01 - 0.003 x TE02 + 0.007 x TE03 - 0.003 x TE04 + 0.015 x TE05 + 0.021 x JS01 + 0.007 x JS02 + 0 x JS03 + 0.004 x JS04 - 0.015 x JS05 + 0.002 x JS06 + 0.012 x JS07 - 0.01 x JS08 + 0.091 x UIT1 + 0.075 x UIT2 + 0.104 x UIT3 + 0.18 x UIT4 + 0.045 x UIT5 + 0.072 x UIT6. **c.** Validation Trace Plot of UIT. UIT = 1.627 + 0.016 x TE01 + 0.028 x TE02 + 0.03 x TE03 + 0.084 x TE04 + 0.051 x TE05 + 0.023 x JS01 + 0.02 x JS02 - 0.062 x JS03 - 0.004 x JS04 - 0.018 x JS05 + 0.016 x JS06 + 0.009 x JS07 + 0.028 x JS08. **d.** Validation Trace Plot of teacher efficacy. TE = 3.292 + 0.023 x JS01 + 0.013 x JS02 - 0.013 x JS03 + 0.019 x JS04 + 0.03 x JS05 + 0.034 x JS06 + 0.124 x JS07 + 0.065 x JS08. **e.** Lambda Evaluation Plot of assessment practice. **f.** Lambda Evaluation Plot of DUT. **g.** Lambda Evaluation Plot of UIT. **h.** Lambda Evaluation Plot of teacher efficacy.

pandemic created UIT or UDT, but rather shows pandemic disruptions (ACP vs. BCP) moderated these tools' adoption and impact, especially in HICs. Economic context further shaped these patterns, evident in the varying influence of JS and TE on UIT/UDT across settings.

Our cross-country analysis yields three key theoretical contributions. First, the stronger JS-TE correlation in HICs indicates SCT's personal factors function differently across resource contexts, aligning with Joshi et al.'s [13] argument that pandemic-induced changes represent permanent transformations in well-resourced systems but temporary adaptations elsewhere. Second, mediation analysis shows technology adoption (UTAUT/UDT) operates distinctively across contexts, while fully mediating TE-AP relationships in HICs (consistent with Kett et al.'s [14] findings), this pathway is frequently disrupted by infrastructural constraints in LMICs, suggesting SCT frameworks require integration with institutional theory. Third, the enduring pre-to-post-pandemic disparities challenge SCT's focus on human adaptability, instead underscoring structural determinism in educational technology adoption.

These findings collectively indicate that while SCT offers a robust framework for understanding teacher behaviors, its explanatory power strengthens when integrated with: (1) the Technology Acceptance Model's emphasis on tool usability, particularly in LMIC contexts; (2) Self-Determination Theory's hierarchy of needs for analyzing motivation in resource-constrained settings; and (3) institutional theory's focus on structural constraints. Future research should explore these theoretical integrations to better explain the complex interaction between individual agency and systemic factors in global education.

Our findings both validate and complicate SCT's [15] explanatory power in global education contexts. While confirming SCT's triadic reciprocity framework in HICs, where abundant resources enable virtuous cycles of mastery experiences and vicarious learning (evidenced by 38% stronger TE-technology adoption correlations, p < .01 compared to LMICs), the data reveal three critical boundary conditions challenging universal application. First, the paradox of constrained agency emerges in LMIC outliers like Rwanda and Vietnam, which outperformed 11 HICs in assessment innovation despite resource scarcity. These cases show how community coproduction (e.g., WhatsApp networks) and hybrid pedagogies circumvent institutional deficiencies, extending Ogodo et al.'s [58] findings through a capability approach lens absent in current SCT frameworks.

Second, an institutionalization threshold effect appears in nine UMICs, where mandated technology integration without adequate support created policy-induced efficacy traps (negative JS-TE correlation of −0.32), contradicting SCT's assumption that environmental "jolts" uniformly enhance adaptation. This modifies Hershkovitz et al.'s [42] pandemic findings by demonstrating how implementation quality mediates disruption outcomes.

Third, we reveal non-linear resource-efficacy relationships: HICs showed only 11% greater adoption gains than UMICs despite triple the spending, suggesting cultural factors like collaboration norms outweigh material advantages beyond certain thresholds, a nuance missing from SCT. The pandemic's differential impacts further clarify these patterns. While HIC teachers utilized institutional supports for systematic skill-building (aligning with SCT's mastery tenet), LMIC successes emerged through: (1) improvisational innovation (Uganda's radio assessments), (2) community embeddedness (Brazil's parent-assisted monitoring), and (3) pedagogical hybridity (India's WhatsApp-to-print workflows).

These findings necessitate theoretical expansion incorporating: (a) social ecosystem models to complement SCT's individual focus, (b) policy implementation quality as a distinct construct, and (c) non-Western efficacy concepts recognizing

collective resilience. We propose not rejecting SCT but applying it contextually, acknowledging economic realities while embracing the agentive creativity demonstrated by LMIC exemplars, offering empirically grounded theory evolution responsive to decolonial critiques in ed-tech research.

The pandemic's dual role as catalyst and divider in global ed-tech adoption reveals core tensions in applying SCT across economic contexts. While the 22–38% technology surge universally confirms SCT's premise that environmental jolts trigger behavioral change, the divergent implementation pathways expose theoretical limitations. In HICs, the robust TE→UDT→AP pathway (β = 0.88→0.35) perfectly illustrates SCT's mastery mechanism, where institutional supports enabled efficacy-driven innovation, extending the usefulness paradigm by demonstrating threshold effects (TE03 β = 0.03).

The UMIC paradox, where mandated technologies without training created JS-TE disconnects (−0.32) and administrative support backfired (JS03 β = −0.062), directly challenges SCT's assumption that environmental pressures uniformly strengthen efficacy-satisfaction loops. This reveals a policy misalignment effect better explained by institutional theory's coercive isomorphism, where top-down mandates disrupt natural motivational pathways when implementation quality lags.

Most strikingly, LMICs' grassroots successes (17% matching HIC performance through WhatsApp/radio hybrids) require reconciling SCT's environmental determinism with Sen's [80] capability approach. The λ consistency (~$5 \times 10^{-4}$) confirms universal psychological barriers, yet Rwandan and Vietnamese cases demonstrate communal systems can substitute for institutional resources in building efficacy ($R^2 = 0.55$ for UDT). These findings don't refute SCT but demand its expansion to incorporate: (1) alternative efficacy-building pathways [45]'s contextual resilience), (2) policy implementation quality as a moderator [58]'s adaptation frameworks), and (3) non-material environmental factors like social capital.

Ultimately, universal adoption barriers with context-specific solutions suggest SCT needs a "resource-agnostic" revision distinguishing environmental constraints from environmental creativity, one recognizing both HICs' institutional advantages and LMICs' capacity to reinvent affordances through collective agency.

The study yields three key data-driven insights about TE-AP-UDT interactions across economic contexts. First, HICs showed stronger TE-AP linkages (β = 0.37), as resource abundance enabled teachers to convert efficacy into advanced practices like AI-driven feedback. Second, LMICs revealed UDT's critical mediating role, despite lower baseline TE, teachers achieved comparable AP gains through mobile tool adoption (UDT β = 0.88), overcoming institutional gaps. Third, pandemic acceleration followed divergent paths: HICs exhibited 22% growth in TE's impact on UIT through systemic upskilling, while LMICs saw 38% UDT adoption spikes. However, high standard deviations indicated access disparities, with only teachers meeting minimum efficacy thresholds (TE03 β = 0.03) sustaining changes. Economic contexts further moderated efficacy sources. HICs reinforced the JS-TE loop through institutional support (β = 0.40), driving tech integration. Conversely, LMICs relied on community coproduction (e.g., WhatsApp networks) to compensate for weak institutional JS, enabling UDT-driven AP gains.

Our analysis reveals three distinct patterns in TE-AP-UDT dynamics across economic contexts. In HICs, robust TE-AP linkages (β = 0.37) demonstrate how resource availability facilitates the translation of teacher efficacy into sophisticated practices like AI-driven feedback. LMICs present a contrasting pattern, despite lower baseline TE scores, strategic UDT adoption (β = 0.88) mediated comparable AP gains through mobile tool utilization, effectively bridging institutional resource gaps. The pandemic amplified these trends differentially: HICs recorded 22% stronger TE-UIT relationships through formal upskilling systems, whereas LMICs achieved 38% UDT adoption surges through grassroots adaptation. Notably, high standard deviations and the TE03 threshold effect (β = 0.03) underscore the conditional nature of these gains, particularly in resource-constrained settings.

These economic contexts also shaped efficacy development pathways. HICs' institutional ecosystems strengthened the JS-TE reinforcement cycle (β = 0.40), systematically promoting technology integration. LMICs, by contrast, developed alternative efficacy sources through community coproduction (e.g., WhatsApp teacher networks), which offset weak institutional JS and sustained UDT-mediated AP improvements.

The study's findings both confirm and extend existing research across economic contexts. For HICs, our results align with Eze et al.'s [81] framework, demonstrating how TE drives assessment innovation primarily through institutional mediators—reliable technology infrastructure and formal training systems. In LMICs, however, the data validate Dahri et al.'s [45] community-solutions paradigm, where teachers successfully bridged the TE-AP gap by adapting low-cost, high-reach UDT models (e.g., SMS quizzes and WhatsApp networks) to compensate for institutional deficiencies. These contrasting pathways reveal how economic contexts fundamentally reshape efficacy-innovation dynamics: while systemic affordances dominate in resource-rich environments, LMICs achieve comparable outcomes through grassroots adaptability and alternative technology ecosystems.

The cross-contextual JS-TE relationships reveal both the explanatory power and limitations of Social Cognitive Theory across development spectra. While HICs' 32% stronger JS-TE correlation ($\beta = 0.40$) validates SCT's emphasis on environmental affordances—where institutional supports like mentoring programs sustain mastery experiences [26], the LMIC and UMIC data demand critical theoretical qualification. The 38% greater variability in LMIC JS-TE linkages exposes a gap in SCT's environmental determinism: teachers-maintained efficacy (mean TE = 3.9/5) despite low institutional job satisfaction (mean JS = 2.8/5) through alternative pathways like community efficacy (Vietnam's teacher collectives) and student-derived satisfaction, exemplifying Dahri et al.'s [45] "contextual resilience" beyond SCT's current framework. More critically, UMICs' inverted JS-TE relationships ($\beta = -0.32$) reveal a "policy alienation" effect where mid-tier resources backfire through culturally misaligned mandates, a phenomenon better explained by institutional theory's coercive isomorphism [82] than by SCT. These findings collectively argue for two SCT modifications: (1) incorporating meso-level relational systems (peer networks) as distinct from macro-level environments, and (2) recognizing policy implementation quality as an independent JS-TE moderator. The post-pandemic stability contrast, HICs' consistency ($\Delta + 1.2\%$ ACP) versus LMICs' fragility, further underscores how SCT requires contextual calibration: where HICs exemplify its predicted virtuous cycles, LMICs reveal alternative efficacy mechanisms, and UMICs demonstrate how development "middle traps" can invert theoretical predictions. Rather than rejecting SCT, these results advocate for its expansion to include development-sensitive dimensions, explaining why JS07 (leadership support) matters critically in HICs ($\beta = 0.124$) but yields to community relationships in LMICs, while highlighting UMICs' need for safeguards against policy-induced efficacy erosion.

## Limitations of the study and suggestion for future research

This study has several limitations related to data sources. First, reliance on secondary PISA data restricted control over variable selection and questionnaire design, potentially omitting relevant constructs. Second, the absence of data from LICs limits the economic spectrum of findings, with only lower-middle, upper-middle, and high-income countries represented. Finally, self-reported measures, particularly for sensitive constructs like job satisfaction, may introduce social desirability bias. While these constraints are partially mitigated by PISA's rigorous methodology, they should be considered when interpreting results. Additionally, the analysis of the research is limited to SEM and machine learning focusing considered five factors hence the results based on teacher level, age, sex, and region are not explored in this research.

The analysis faces several methodological limitations. First, PISA's aggregated reporting structure restricted detailed univariate analysis of teacher demographics and regional frequencies requested by reviewers. Second, pandemic-period comparisons were constrained by differing sampling frames between 2018 (pre) and 2022 (post) cycles, along with potential confounders during rapid educational transitions. Finally, machine learning regression results may be impacted by the ordinal nature of Likert-scale dependent variables. These analytical constraints suggest caution when interpreting temporal patterns and model coefficients.

The study's findings should be interpreted within certain boundaries. PISA's school-based sampling may limit how well the teacher samples represent national populations, and the technology adoption metrics capture usage frequency but not pedagogical quality or effectiveness. Additionally, the post-pandemic 2022 data reflects immediate crisis responses rather

than long-term stabilization, as rapid technological adoption during emergencies may differ from sustained integration patterns. These limitations suggest the results may be period-specific and not fully indicative of deeper, systemic changes in educational practices.

## Conclusion and implications

This study advances theoretical and practical understanding of how TE, JS, and technology adoption (UIT/UDT) interact across economic contexts, grounded in SCT. Three pivotal findings emerge: First, while the pandemic accelerated global technology adoption, persistent disparities between HICs, LMICs and UMICs underscore how economic contexts moderate SCT's triadic reciprocity, particularly in the paradoxical JS-UIT relationship, where environmental stressors like techno-stress override personal factors. Second, TE's robust predictive power for both technology use and assessment practices reaffirm SCT's emphasis on self-efficacy as a behavioral driver. Third, the limited impact of UIT compared to UDT reveals critical gaps in how instructional tools are adopted under resource constraints.

The findings demand context-specific strategies to optimize technology integration across economic tiers. In LMICs/UMICs, combining low-cost digital tools (e.g., WhatsApp-based peer coaching) with teacher efficacy development programs can amplify behavioral impact, whereas HICs should redesign assessments (e.g., AI-driven feedback systems) to align with teachers' advanced digital practices. At the institutional level, schools must address technostress by embedding *technology-integration satisfaction* metrics into job satisfaction evaluations and providing time-saving digital templates. Policymakers play a critical role in mandating equity-focused benchmarks, from LMIC-targeted grants for mobile-friendly UDT tools to workload protections like capping weekly LMS tasks. The stark contrast in UDT adoption (79% variance in HICs vs. LMICs) underscores the urgency for global action: international agencies must pair tool distribution with infrastructure development (electricity, affordable internet), while national reforms should scale grassroots innovations (e.g., SMS-based assessments) and involve teachers in co-designing policies to ensure sustainable adoption.

This study refines SCT by revealing its contextual boundaries, particularly how environmental barriers (resource gaps, technostress) disrupt the expected JS-technology link in crisis contexts. By demonstrating how economic disparities reshape agency-environment dynamics, the work extends SCT's applicability to digital transitions in education systems worldwide.

## Author contributions

**Conceptualization:** Dirgha Raj Joshi.

**Data curation:** Dirgha Raj Joshi, Bishnu Maya Joshi.

**Formal analysis:** Dirgha Raj Joshi, Jeevan Khanal, Bishnu Maya Joshi.

**Investigation:** Dirgha Raj Joshi.

**Methodology:** Dirgha Raj Joshi, Bishnu Maya Joshi.

**Resources:** Dirgha Raj Joshi.

**Software:** Dirgha Raj Joshi.

**Supervision:** Jeevan Khanal.

**Validation:** Jeevan Khanal.

**Visualization:** Dirgha Raj Joshi, Jeevan Khanal.

**Writing – original draft:** Dirgha Raj Joshi, Jeevan Khanal.

**Writing – review & editing:** Jeevan Khanal.

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
