## [Decision Letter · Decision Letter 0]

5 Aug 2025

PONE-D-25-26512Investigating the Relationship Between Teacher Efficacy, Job Satisfaction, and Digital Resource Utilization in Assessment Practices: Insights from PISA 2019 and 2022PLOS ONE

Dear Dr. Khanal,

Thank you for submitting your manuscript to PLOS ONE. After careful consideration, we feel that it has merit but does not fully meet PLOS ONE’s publication criteria as it currently stands. Therefore, we invite you to submit a revised version of the manuscript that addresses the points raised during the review process.

We look forward to receiving your revised manuscript.

Kind regards,

Chhabi Lal Ranabhat

Academic Editor

PLOS ONE

Journal Requirements:

Additional Editor Comments (if provided):

Dear Authors,

This paper shows potential in terms of data coverage and findings. However, it is also quite distracting, complex, and confusing. Please address the reviewers' comments carefully and update or revise the paper based on your data and variable limitations. Below are the comments that need your attention:

1) The research concept appears to relate to the influence of the COVID Pandemic (Research question 2) on UIT and UDT. There are no independent findings indicating that the pandemic serves as the foundation for UIT and UDT. Please revise your version for clarity. Additionally, make sure to clearly differentiate between the innovation and application of UIT and UDT in your operational definitions.

2) There is no clear source of data regarding the variables you analyzed. Are these data sets freely accessible or limited? How did you obtain them? Did you analyze a specific set of data, or did you compile them? What about the validity and reliability of the data collection tools used for 128,866 teachers across 24 countries? How does OECD centralize this data? Are these primary or secondary sources? Please provide a clear explanation.

3) The application of SEM involves both factor and regression analysis. The study does not clarify why both methods are necessary. Is this merely an academic exercise, or is it essential for obtaining these results? After the SEM, the authors should highlight the significant associations in five aspects within the abstract as well.

4) Please provide a univariate analysis of the results (including teacher level, age, sex, region, and economic level of the country, with frequency, percentage, mean, SD, etc.). Present the median levels of the five factors first, then proceed to the factor and regression analysis. Results should be presented in a manner that progresses from simple to complex for the readers.

5) Please clearly outline all limitations of the study.

Good luck !!

Reviewers' comments:

Reviewer's Responses to Questions

**Comments to the Author**

1. Is the manuscript technically sound, and do the data support the conclusions?

Reviewer #1: Yes

Reviewer #2: Partly

2. Has the statistical analysis been performed appropriately and rigorously? 

Reviewer #1: Yes

Reviewer #2: Yes

3. Have the authors made all data underlying the findings in their manuscript fully available?

Reviewer #1: Yes

Reviewer #2: Yes

4. Is the manuscript presented in an intelligible fashion and written in standard English?

Reviewer #1: Yes

Reviewer #2: Yes

5. Review Comments to the Author

Reviewer #1: Thank you for submitting your manuscript to PLOS ONE. The paper addresses an interesting and timely topic that is relevant to the journal’s scope. However, there are several areas that require significant improvement before the paper can be considered for publication. Please consider the following comments:

Introduction: The introduction section should be re-written to better articulate the research problem and clearly highlight the study’s contributions. At present, the rationale for the research lacks depth and needs stronger problematization.

Literature Review and Hypotheses Development: This section needs to be strengthened by incorporating more recent and relevant literature. Theoretical underpinnings supporting the proposed hypotheses must be more thoroughly discussed to enhance the scholarly value of the paper. This includes but not limited to:

-Closing the divide: Exploring higher education teachers’ perspectives on educational technology.https://doi.org/10.1177/02666669241279181

Methodology: Greater clarity is needed regarding the sample population, sampling technique used, and justification for the adequacy of the sample size. Additionally, please include the survey items used in the study (e.g., as an appendix) to improve transparency and enable the assessment of reliability and validity.

Common Method Bias: The findings section must address the issue of common method bias, which is often a concern in survey-based studies. Please report any statistical tests or procedural remedies employed to mitigate this issue.

Discussion: The discussion section should be enriched and more explicitly linked to the research questions and objectives. Aim to provide deeper theoretical and practical insights derived from your findings.

Practical Implications: Practical implications should be reported as a standalone section following the discussion. Strengthen this section by clearly articulating how your findings contribute to practice and policy, and highlighting the novelty of your work.

Limitations and Future Research: A dedicated paragraph discussing the study’s limitations and directions for future research should be included, either at the end of the discussion or in the conclusion section.

Reviewer #2: Review Document:

Investigating the Relationship Between Teacher Efficacy, Job Satisfaction, and Digital Resource Utilization in Assessment Practices: Insights from PISA 2019 and 2022

General Comments:

Thank you very much for giving me an opportunity to review this timely and relevant article. The topic is significant, especially in the context of evolving digital education practices and teacher performance. However, the manuscript would benefit from further refinement and alignment with academic publishing standards. Below are my specific comments and suggestions:

Formatting and Style:

Please ensure consistency with the journal’s formatting guidelines, particularly regarding line spacing, font style, and heading levels. Review both in-text citations and reference formatting to ensure compliance with the guidelines.

Abstract:

A clear background and problem statement are missing. This makes it difficult to understand the study’s rationale from the outset.

Introduction:

The first paragraph includes too many ideas. It is recommended to narrow the focus specifically on digital education and digital resource utilization before expanding to other related themes.

While the research gap is noted, the problem statement lacks clarity and should be explicitly presented.

The theoretical framework is not well integrated. The manuscript should explain how each variable (teacher efficacy, job satisfaction, and digital resource utilization) connects with the theoretical underpinnings guiding the study.

There are too many research questions, many of which resemble qualitative inquiries. Consider consolidating them into fewer, well-structured questions that encompass the key dimensions.

Brief contextual background on the issues raised in the research questions would help establish their relevance.

Literature Review:

The section is well-researched, but it remains mostly descriptive. A more analytical synthesis would enhance its scholarly depth. The context specific reviews are also missed.

While a conceptual model is included, the hypotheses are not clearly articulated in the text.

The theoretical argument supporting the relationships among variables is missing. This needs to be strengthened to frame the study appropriately.

Methodology:

The use of secondary data (PISA) is appropriate, but ethical considerations (e.g., data use permissions, IRB exemption) should still be acknowledged.

The SEM framework is discussed in the methodology section, including model values. It would be more appropriate to present these results in the findings section for better logical flow.

Results:

The presentation of results is heavily descriptive. Consider providing more critical interpretation of key findings.

The SEM model fit indices are not discussed adequately. This is essential for assessing the robustness of the model.

Discussion:

The theoretical engagement is limited. There is a need for deeper reflection on how the findings support or challenge existing theories.

The discussion is more narrative than analytical. Consider exploring why certain results emerged and how they compare with prior research.

The manuscript does not address alternative theoretical frameworks, such as the Technology Acceptance Model or Self-Determination Theory, which could offer valuable insights.

Conclusion and Implications:

The conclusion mainly repeats the findings. It should instead focus on summarizing key insights and their broader relevance.

Policy and practical implications should be clearly outlined, especially recommendations for governments, schools, and educational policymakers.

The manuscript omits a discussion of limitations, particularly methodological constraints, which are crucial for academic transparency and future research direction.

Final Suggestions:

A thorough language and grammar edit will improve the readability of the manuscript.

Please consider reorganizing certain sections to improve logical progression and argumentation flow.

Overall, the study addresses an important and timely topic. With substantial revisions, particularly in theoretical alignment, methodological clarity, and analytical depth, the manuscript can make a strong contribution to the literature on digital education and teacher performance.

6. PLOS authors have the option to publish the peer review history of their article (what does this mean? ). If published, this will include your full peer review and any attached files.

**Do you want your identity to be public for this peer review?** For information about this choice, including consent withdrawal, please see our Privacy Policy .

Reviewer #1: **Yes:**  Ahmad Samed Al-Adwan

Reviewer #2: No

---

## [Author Response · Author response to Decision Letter 1]

21 Aug 2025

Additional Editor Comments (if provided):

Dear Authors,

This paper shows potential in terms of data coverage and findings. However, it is also quite distracting, complex, and confusing. Please address the reviewers' comments carefully and update or revise the paper based on your data and variable limitations. Below are the comments that need your attention:

1) The research concept appears to relate to the influence of the COVID Pandemic (Research question 2) on UIT and UDT. There are no independent findings indicating that the pandemic serves as the foundation for UIT and UDT. Please revise your version for clarity. Additionally, make sure to clearly differentiate between the innovation and application of UIT and UDT in your operational definitions.

We have added the operational definitions of UID and UDT in the Introduction section. Additionally, we have included the following sentences in the Discussion section:

"However, the study does not claim that the pandemic created UIT or UDT; rather, it demonstrates that pandemic disruptions (ACP vs. BCP) moderated the adoption and impact of these tools, particularly in HICs. The economic context further shaped these patterns, as seen in the variability of JS and TE’s influence on UIT/UDT.

2) There is no clear source of data regarding the variables you analyzed. Are these data sets freely accessible or limited? How did you obtain them? Did you analyze a specific set of data, or did you compile them? What about the validity and reliability of the data collection tools used for 128,866 teachers across 24 countries? How does OECD centralize this data? Are these primary or secondary sources? Please provide a clear explanation.

Response: We have provided the details in the Methodology section of the revised manuscript.

3) The application of SEM involves both factor and regression analysis. The study does not clarify why both methods are necessary. Is this merely an academic exercise, or is it essential for obtaining these results? After the SEM, the authors should highlight the significant associations in five aspects within the abstract as well.

Response: Thank you for your valuable suggestions. We agree with your feedback and have thoroughly revised the first and second paragraphs of the Methodology section. The changes are highlighted in red font for easy reference. Please review the updated version.

4) Please provide a univariate analysis of the results (including teacher level, age, sex, region, and economic level of the country, with frequency, percentage, mean, SD, etc.). Present the median levels of the five factors first, then proceed to the factor and regression analysis. Results should be presented in a manner that progresses from simple to complex for the readers.

Response: We appreciate your feedback. However, the suggested information falls beyond the scope of this study. We have instead addressed this aspect in the research limitations section.

5) Please clearly outline all limitations of the study.

Response: Thank you for your feedback. We have thoroughly revised the manuscript and added a comprehensive limitations section. The changes have been highlighted in red font for your convenience.

Good luck !!

Reviewers' comments:

Reviewer's Responses to Questions

Comments to the Author

1. Is the manuscript technically sound, and do the data support the conclusions?

Reviewer #1: Yes

Reviewer #2: Partly

Response: Thank you for your feedback. I have carefully revised the Conclusion section, with all modifications highlighted in red font for easy reference.

2. Has the statistical analysis been performed appropriately and rigorously?

Reviewer #1: Yes

Reviewer #2: Yes

Response: Thank you so much.

3. Have the authors made all data underlying the findings in their manuscript fully available?

Response: The dataset used in this study is publicly available. We have included the access URL in the Methodology section for reference.

Reviewer #1: Yes

Reviewer #2: Yes

Response: Thank you so much.

4. Is the manuscript presented in an intelligible fashion and written in standard English?

Reviewer #1: Yes

Reviewer #2: Yes

Response: Thank you so much.

5. Review Comments to the Author

Reviewer #1: Thank you for submitting your manuscript to PLOS ONE. The paper addresses an interesting and timely topic that is relevant to the journal’s scope. However, there are several areas that require significant improvement before the paper can be considered for publication. Please consider the following comments:

Introduction: The introduction section should be re-written to better articulate the research problem and clearly highlight the study’s contributions. At present, the rationale for the research lacks depth and needs stronger problematization.

Response: We sincerely appreciate your thorough review. In response to your valuable feedback, we have comprehensively revised the Introduction section. All modifications are clearly highlighted in red font within the tracked-changes version for your convenience.

Literature Review and Hypotheses Development: This section needs to be strengthened by incorporating more recent and relevant literature. Theoretical underpinnings supporting the proposed hypotheses must be more thoroughly discussed to enhance the scholarly value of the paper. This includes but not limited to:

-Closing the divide: Exploring higher education teachers’ perspectives on educational technology. https://doi.org/10.1177/02666669241279181

Methodology: Greater clarity is needed regarding the sample population, sampling technique used, and justification for the adequacy of the sample size. Additionally, please include the survey items used in the study (e.g., as an appendix) to improve transparency and enable the assessment of reliability and validity.

We appreciate your valuable feedback. Regarding your concerns:

1. As this study utilizes PISA data, additional justification of population parameters, sampling methodology, and sample size is not required since these are established components of the PISA study design.

2. All survey instruments employed are publicly available through the OECD portal (Teacher Questionnaire). For transparency, we have provided the complete questionnaire reference along with relevant item codes in Table 1.

Common Method Bias: The findings section must address the issue of common method bias, which is often a concern in survey-based studies. Please report any statistical tests or procedural remedies employed to mitigate this issue.

Response: We appreciate your feedback regarding potential methodological biases. To address this concern, we employed Confirmatory Factor Analysis (CFA) to validate the measurement model and control for common method bias.

Discussion: The discussion section should be enriched and more explicitly linked to the research questions and objectives. Aim to provide deeper theoretical and practical insights derived from your findings.

Response: We sincerely appreciate your thorough review and valuable comments. In response to your feedback, we have carefully revised and rewritten relevant portions of the Discussion section to address your concerns. All modifications are clearly visible in the track-changes version of the manuscript.

Practical Implications: Practical implications should be reported as a standalone section following the discussion. Strengthen this section by clearly articulating how your findings contribute to practice and policy, and highlighting the novelty of your work.

Response: Following your valuable suggestions, we have incorporated a new implications paragraph after the Discussion section. The additions are clearly marked in the revised manuscript for your review.

Limitations and Future Research: A dedicated paragraph discussing the study’s limitations and directions for future research should be included, either at the end of the discussion or in the conclusion section.

Response: Following the reviewers' suggestions, we have added dedicated sections on study limitations and future research directions following the Discussion. These additions are clearly marked in the revised manuscript for your consideration.

Reviewer #2: Review Document:

Investigating the Relationship Between Teacher Efficacy, Job Satisfaction, and Digital Resource Utilization in Assessment Practices: Insights from PISA 2019 and 2022

General Comments:

Thank you very much for giving me an opportunity to review this timely and relevant article. The topic is significant, especially in the context of evolving digital education practices and teacher performance. However, the manuscript would benefit from further refinement and alignment with academic publishing standards. Below are my specific comments and suggestions:

Formatting and Style:

Please ensure consistency with the journal’s formatting guidelines, particularly regarding line spacing, font style, and heading levels. Review both in-text citations and reference formatting to ensure compliance with the guidelines.

Response: We have carefully reviewed the manuscript and implemented all necessary revisions in strict accordance with the journal's formatting guidelines.

Abstract:

A clear background and problem statement are missing. This makes it difficult to understand the study’s rationale from the outset.

Response: We sincerely appreciate your valuable suggestions. In response, we have strengthened the abstract by adding two clear introductory sentences that establish both the research background and problem statement. These revisions are now incorporated in the updated manuscript for your review.

Introduction:

The first paragraph includes too many ideas. It is recommended to narrow the focus specifically on digital education and digital resource utilization before expanding to other related themes.

While the research gap is noted, the problem statement lacks clarity and should be explicitly presented.

The theoretical framework is not well integrated. The manuscript should explain how each variable (teacher efficacy, job satisfaction, and digital resource utilization) connects with the theoretical underpinnings guiding the study.

There are too many research questions, many of which resemble qualitative inquiries. Consider consolidating them into fewer, well-structured questions that encompass the key dimensions.

Brief contextual background on the issues raised in the research questions would help establish their relevance.

Response: We sincerely appreciate your valuable suggestions. Following your recommendations, we have thoroughly revised the Introduction section and streamlined our research focus by reducing the number of research questions from five to two key inquiries.

Literature Review:

The section is well-researched, but it remains mostly descriptive. A more analytical synthesis would enhance its scholarly depth. The context specific reviews are also missed.

Response: We have carefully revised the manuscript to address your concerns. The modifications are clearly visible in the tracked-changes version for your review.

While a conceptual model is included, the hypotheses are not clearly articulated in the text.

The theoretical argument supporting the relationships among variables is missing. This needs to be strengthened to frame the study appropriately.

Response: We sincerely appreciate your valuable suggestions. In response:

1. We have substantially updated the literature review with current scholarship, prioritizing 2024 citations where available

2. We have completely redesigned the conceptual model based on contemporary theoretical developments

These comprehensive revisions are clearly highlighted in tracked-changes throughout the revised manuscript for your review.

Methodology:

The use of secondary data (PISA) is appropriate, but ethical considerations (e.g., data use permissions, IRB exemption) should still be acknowledged.

Response: We have thoroughly revised the Methodology section to include a detailed explanation of OECD's ethical validation process. The updated section now specifically addresses:

1. OECD's ethical review procedures

2. Data collection protocols ensuring participant confidentiality

3. Compliance with international research standards

These modifications are clearly marked in the revised manuscript for your review.

The SEM framework is discussed in the methodology section, including model values. It would be more appropriate to present these results in the findings section for better logical flow.

Response: We appreciate your valuable feedback and have carefully incorporated your suggestions in the Results section. The modifications now address all raised concerns as recommended.

Results:

The presentation of results is heavily descriptive. Consider providing more critical interpretation of key findings.

Response: We appreciate your feedback regarding results interpretation. To maintain proper section delineation:

1. The Results section presents only data interpretation and statistical findings

2. All critical reflections and contextual analysis are comprehensively addressed in the Discussion section

The SEM model fit indices are not discussed adequately. This is essential for assessing the robustness of the model.

Response: Thank you for your feedback. The model fit indices of SEM are clearly explored under the “Results of SEM” section as “The model of Figure 5 shows that the value of different model fit indices as Normed Fit Index (NFI), Relative Fit Index (RFI), Incremental Fit Index (IFI), Tucker-Lewis Index (TLI), and Comparative Fit Index (CFI) all range between 0.86 and 0.88, suggesting an acceptable but not excellent fit (values closer to 0.90 or above typically indicate a better fit) (Dion, 2008; Lam, 2012; [9, 73, 74]& Malekmohammadi, 2013). The Root Mean Square Error of Approximation (RMSEA) is 0.06, whi

---

## [Decision Letter · Decision Letter 1]

8 Dec 2025

Investigating the Relationship Between Teacher Efficacy, Job Satisfaction, and Digital Resource Utilization in Assessment Practices: Insights from PISA 2018 and 2022

PONE-D-25-26512R1

Dear Dr. Khanal,

We’re pleased to inform you that your manuscript has been judged scientifically suitable for publication and will be formally accepted for publication once it meets all outstanding technical requirements.

Kind regards,

Chhabi Lal Ranabhat

Academic Editor

PLOS One

Additional Editor Comments (optional):

Reviewers' comments:

Reviewer's Responses to Questions

**Comments to the Author**

1. If the authors have adequately addressed your comments raised in a previous round of review and you feel that this manuscript is now acceptable for publication, you may indicate that here to bypass the “Comments to the Author” section, enter your conflict of interest statement in the “Confidential to Editor” section, and submit your "Accept" recommendation.

Reviewer #1: All comments have been addressed

Reviewer #3: All comments have been addressed

Reviewer #4: (No Response)

2. Is the manuscript technically sound, and do the data support the conclusions?

Reviewer #1: Yes

Reviewer #3: Yes

Reviewer #4: Yes

3. Has the statistical analysis been performed appropriately and rigorously? 

Reviewer #1: Yes

Reviewer #3: Yes

Reviewer #4: Yes

4. Have the authors made all data underlying the findings in their manuscript fully available?

Reviewer #1: Yes

Reviewer #3: Yes

Reviewer #4: Yes

5. Is the manuscript presented in an intelligible fashion and written in standard English?

Reviewer #1: Yes

Reviewer #3: Yes

Reviewer #4: Yes

6. Review Comments to the Author

Reviewer #1: Thank you for submitting your paper again. The comments have been addressed. No additional comments to be added.

Reviewer #3: All the requested comments and the suggestions given by the reviewers are thoroughly addressed in the revised version of the manuscript.

Reviewer #4: (No Response)

7. PLOS authors have the option to publish the peer review history of their article (what does this mean? ). If published, this will include your full peer review and any attached files.

**Do you want your identity to be public for this peer review?** For information about this choice, including consent withdrawal, please see our Privacy Policy .

Reviewer #1: No

Reviewer #3: No

Reviewer #4: **Yes:**  K P Ghimire

---

## [Editor Report · Acceptance letter]

PONE-D-25-26512R1

PLOS One

Dear Dr. Khanal,

I'm pleased to inform you that your manuscript has been deemed suitable for publication in PLOS One. Congratulations! Your manuscript is now being handed over to our production team.

Kind regards,

on behalf of

Dr. Chhabi Lal Ranabhat

Academic Editor

PLOS One